# Socioeconomic Status and Distance to Reference Centers for Complex Cancer Diseases: A Source of Health Inequalities? A Population Cohort Study Based on Catalonia (Spain)

**DOI:** 10.3390/ijerph19148814

**Published:** 2022-07-20

**Authors:** Paula Manchon-Walsh, Luisa Aliste, Josep M. Borràs, Cristina Coll-Ortega, Joan Casacuberta, Cristina Casanovas-Guitart, Montse Clèries, Sergi Cruz, Àlex Guarga, Anna Mompart, Antoni Planella, Alfonso Pozuelo, Isabel Ticó, Emili Vela, Joan Prades

**Affiliations:** 1Catalonian Cancer Strategy, Department of Health, Government of Catalonia, Avenida Gran Via de l’Hospitalet, 199-203, 08908 L’Hospitalet de Llobregat, Spain; laliste@iconcologia.net (L.A.); jmborras@iconcologia.net (J.M.B.); cristina.coll.ortega@gmail.com (C.C.-O.); jprades@iconcologia.net (J.P.); 2Biomedical Research Institute of Bellvitge (IDIBELL), University of Barcelona, C/Feixa Llarga, s/n, 08907 L’Hospitalet de Llobregat, Spain; 3Cartographic and Geological Institute of Catalonia, Parc de Montjuïc, 08038 Barcelona, Spain; joan.casacuberta@icgc.cat (J.C.); isabel.tico@icgc.cat (I.T.); 4Health Service Procurement and Assessment, Catalonian Health Service (CatSalut), Government of Catalonia, Travessera de les Corts, 131-159, 08028 Barcelona, Spain; casanovas.cris@catsalut.cat (C.C.-G.); aguarga@catsalut.cat (À.G.); apozuelo@catsalut.cat (A.P.); 5Healthcare Information and Knowledge Unit, Department of Health, Government of Catalonia, Gran Via de les Corts Catalanes, 591, 08007 Barcelona, Spain; mcleries@catsalut.cat (M.C.); evela@catsalut.cat (E.V.); 6Digitalization for the Sustainability of the Healthcare System (DS3), Sistema de Salut de Catalunya, Government of Catalonia, Gran Via de les Corts Catalanes, 591, 08007 Barcelona, Spain; 7Subdirectorate-General for the Service Portfolio and Health Map, Directorate-General for Health Planning, Department of Health, Government of Catalonia, Travessera de les Corts, 131-159, 08028 Barcelona, Spain; sergi.cruz@gencat.cat (S.C.); amompart@gencat.cat (A.M.); antoni.planella@gencat.cat (A.P.)

**Keywords:** health equity, centralization, quality of cancer care, health care access, geographical distance, reference centre, socioeconomic status

## Abstract

The centralization of complex surgical procedures for cancer in Catalonia may have led to geographical and socioeconomic inequities. In this population-based cohort study, we assessed the impacts of these two factors on 5-year survival and quality of care in patients undergoing surgery for rectal cancer (2011–12) and pancreatic cancer (2012–15) in public centers, adjusting for age, comorbidity, and tumor stage. We used data on the geographical distance between the patients’ homes and their reference centers, clinical patient and treatment data, income category, and data from the patients’ district hospitals. A composite ‘textbook outcome’ was created from five subindicators of hospitalization. We included 646 cases of pancreatic cancer (12 centers) and 1416 of rectal cancer (26 centers). Distance had no impact on survival for pancreatic cancer patients and was not related to worse survival in rectal cancer. Compared to patients with medium–high income, the risk of death was higher in low-income patients with pancreatic cancer (hazard ratio (HR) 1.46, 95% confidence interval (CI) 1.15–1.86) and very-low-income patients with rectal cancer (HR 5.14, 95% CI 3.51–7.52). Centralization was not associated with worse health outcomes in geographically dispersed patients, including for survival. However, income level remained a significant determinant of survival.

## 1. Introduction

According to the Institute of Medicine, “health equity implies providing care of the same quality to all people, regardless of gender, ethnicity, geographic location, socioeconomic status, or other personal characteristics” [1]. The reference to the distance that separates patients from health services is apt, as geographical factors have generated situations of health inequity among patients, especially those with complex or low-incidence cancers that require treatment in specialized centers [2,3].

However, distance is a concept that must be understood in relative rather than absolute terms. For example, the distance to an expert center may be modulated by the family support that patients have or by the views of professionals from non-expert centers on the advisability of referring patients with advanced age or high comorbidity to distant treatment centers. Given that the mechanisms that would hypothetically explain how geographic factors impact cancer survival are complex and multidimensional [4], they must be specifically evaluated in each health system according to the territorial distribution of the population and the reference centers.

These issues are subject to active debate in the context of cancer care policies involving the centralization of highly complex therapies [5,6,7]. Such policies aim to reduce clinical practice variability and differences in quality that are not justified by the clinical situation or tumor subtype [8]. Numerous European strategies indicate the need for a limited number of specialized, multidisciplinary teams to take on cases of pathologies such as pancreatic cancer [9] or sarcoma [10], as well as other more common ones, such as breast cancer or other tumours with extensive care and resource requirements [11]. Such care scenarios increase the distance that many patients have to travel to reach reference centers and units, so the sought-after improvement in effectiveness may be countered by a negative impact on equity of access. In fact, different studies have shown how geographical distance can impact the probability of undergoing a full range of cancer treatments and the chances of survival [12,13,14].

Catalonia (Spain) began to implement a centralization policy for 20 procedures and highly complex cancer diseases in 2012 [15]. The legislation underpinning this policy [16] (approved in 2012 and updated in 2018) identified and authorized the so-called reference centers, which had to meet minimum annual thresholds of patients treated with a curative intent (e.g., 50 cases for lung cancer, 11 cases for rectal cancer, and 10 cases for pancreatic cancer). All other hospitals were obliged to refer their patients to the nearest reference center to receive care [15]. For sarcoma, this policy led to a consolidation of service provision from 20 centers to 4, and for esophageal cancer surgery, from 18 to 5.

However, certain characteristics of the reform and of the health system itself have given rise to potential equity problems. From the outset, the creation of reference centers was not accompanied by specific time targets for transitioning patients between hospitals or specific regulations on the roles of non-expert centers in diagnosis and tumor staging. Likewise, the policy did not establish the need to have a shared, interhospital clinical protocol, for example, specifying which center should manage postsurgical complications and how; these details were left to the centers to organize themselves. Moreover, existing patient care financing mechanisms (e.g., for travel and lodging) [17] were not related to the reform and were limited in scope. The referral process itself was critical due to potential problems related to the duplication of radiological tests or the interoperability of computer systems, which could end up causing delays and worsening tumor stage at diagnosis [18].

Taking these considerations as a starting point, the hypothesis of this study is that the centralization of highly complex surgical procedures in Catalonia led to inequities based on the distance between patients’ homes and reference centers, as well as their socioeconomic positions. Thus, our aim is to assess the distance from the patients’ homes to the reference centers and their socioeconomic statuses in relation to survival, adjusting for factors such as age, comorbidity, and tumor stage at diagnosis in people with centralized oncological pathologies in Catalonia.

## 2. Methods

### 2.1. Healthcare Setting

The study took place in Catalonia, Spain (pop. 7.7 million), which follows a national health service model characterized by a purchaser–provider split. The public agency (CatSalut) plays a key role in contracting health services from hospitals. Within the comprehensive healthcare system of Catalonia (SISCAT), there are 68 publicly funded centers that deliver hospital care and participate to some extent in cancer care. Eleven of these have radiotherapy units, and three have satellite services (i.e., linear accelerators controlled by the former). About 10% to 15% of cancer procedures are delivered by private providers. Catalonia is a small (31,895 km^2^), densely populated region (242.26 inhabitants/km²), with metropolitan Barcelona (with 1937.4 inhabitants/km^2^) standing out from the rest, including the Pyrenees mountain range.

The centralization decree [16] reorganized the field of highly complex cancer care, giving reference centers the exclusive competency to perform certain surgical interventions, such as rectal or pancreatic cancer surgery, and to assume responsibility for the whole care process—including diagnosis, therapy, and follow-up—for some specific pathologies and conditions, such as pediatric oncology (restricted to two reference centers). Centralization is based on a model combining the accreditation of centers with an external quality assessment system based on clinical audits [19,20]. The results of the clinical audits—implemented by the Catalan Cancer Plan and CatSalut—have led to the revocation of reference center status for some hospitals.

### 2.2. Study Design

We designed a population-based cohort study in patients treated with a curative intent using four types of related data sources: (1) data on the geographical distance between the patients’ homes and their reference center, calculated by the Cartographic and Geological Institute of Catalonia; (2) clinical patient and treatment data from clinical audits (one in rectal cancer and two in pancreatic cancer), including the surgical center; (3) patients’ income categories, based on the pharmacy copayment level (this categorization is determined according to declared income on individual tax returns); and (4) data from the patients’ district hospitals in cases where this center was not a reference center in pancreatic or rectal cancer.

We focused on rectal and pancreatic cancer because of two related factors. First, these pathologies present very different degrees of centralization, with a reduction from 20 to 12 centers for pancreatic cancer and from 51 to 27 centers for rectal cancer. Secondly, surgery for pancreatic cancer is one of the most technically complex and risky surgical interventions performed, while surgery for rectal cancer is a common practice but is considered at the lower limit of a “high complexity” procedure in oncology [9].

### 2.3. Population

All patients undergoing surgery with curative intent for rectal cancer (study period 2011–12) and pancreatic cancer (2012–15) in SISCAT centers were eligible. These patients were included in clinical audits carried out within the framework of the Catalan Cancer Plan and CatSalut’s reorganization strategy for tertiary care, which proposed the centralization of care for oncological pathologies and highly complex procedures in reference centers [2]. In order to specifically assess the impact of distance on survival, we excluded patients operated on in centers that were not reference centers or in reference centers that did not correspond to their health areas according to residence.

### 2.4. Variables

The data collected were: tumor stage based on the 7th edition of the *TNM Classification of Malignant Tumours*; physical status based on the American Society of Anaesthesiologists (ASA) physical status score; reference center; current vital status; distance from home to reference center; individual income based on pharmaceutical copayment category; and existence of a district hospital other than the reference center providing care for rectal and pancreatic cancer. In addition, we created a dichotomous, composite endpoint, *textbook outcome*, with the aim of summarizing the quality of care received in a single indicator (for both pathologies). We considered patients to have a textbook outcome if they met five conditions selected by the research team based on the existing literature [21,22]:Absence of emergency admission;Length of hospital stay not exceeding the 75th percentile;No reintervention within 90 days of the first surgical treatment;Lack of postsurgical complication within 30 days of the first surgical treatment;Radical resection (pancreas: R0; rectum: total mesorectal excision).

Socioeconomic status (SES) was considered an exposure of interest, as proxied by an indicator of annual individual income (tax return) and receipt of welfare assistance. The Catalan Health Surveillance system database collects and updates data on annual income. Four individual-level SES categories were defined: “high SES” (annual income > EUR 100,000 /year); “medium SES” (EUR 18,000–100,000/year); “low SES” (<EUR 18,000/year); and “very low SES” (individuals receiving welfare support from the government). These categories were based on the income groups that determine drug copayments nationally (due to the small count of “high SES” for the analysis, we decided to pool the high and medium SES categories). Both working-age and retired individuals were included in this classification [23].

### 2.5. Statistical Analyses

The main clinical, socioeconomic, and geographical characteristics of the included patients were expressed as medians and interquartile ranges (IQRs) in the case of quantitative data and as numbers of patients and percentages in the case of qualitative data. The Cox proportional hazards model was used to calculate hazard ratios (HR) with 95% confidence intervals (CIs) in the uni- and multivariate analyses of mortality at 5 years. Time to death was assessed using Kaplan–Meier curves. A log-rank test was used to compare mortality between the study groups.

Finally, we performed a subanalysis of the 237 cases of rectal cancer and the 168 cases of pancreatic cancer operated on in authorized centers outside the patients’ health districts.

Statistical analyses were performed using SPSS software (version 21), while QGIS software (QGIS, Grüt, Swiss) was used to calculate the distance from the patients’ homes to the corresponding reference centers according to the centralization strategy. This software also enabled a graphical representation of the distribution of distances in the region. This study was approved by the institutional ethics committee at Bellvitge (PR177/18). It was funded by the Carlos III Health Institute (ISCIII) in the 2019 call for Health Research Projects (PI18/01835).

## 3. Results

A total of 2819 patients were initially deemed eligible. Figure 1 presents a patient flow chart and details the causes of exclusion. Altogether, 25% of the cases included in the audits were treated in authorized centers outside the patients’ health districts or in unauthorized centers. The final sample included 2101 patients: 646 with pancreatic cancer and 1455 with rectal cancer.

Table 1 describes patient characteristics according to the type of pathology. Most patients lived less than 10 km from the center where they were operated on. This percentage was higher in patients with rectal cancer (71.5%) compared to pancreatic cancer (58.8%). Regarding SES, the largest category was low SES, with 1468 patients (69.9%). The proportion of cases with very low SES was higher in rectal (4.8%) compared to pancreatic cancer (2.9%). Moreover, a higher proportion of patients with low income were women (pancreatic cancer: 81.3% vs. 64.6%, *p* < 0.001; rectal: 75.8% vs. 65.6%; *p* < 0.001), along with a lower proportion of patients with middle–high income (pancreatic cancer: 15% vs. 33%, *p* < 0.001; rectal: 18.9% vs. 29.9%, *p* < 0.001; Appendix A).

Surgery for pancreatic cancer took place in 12 centers and, for rectal cancer, in 27, reflecting differences in the centralization strategies between pathologies that condition the distance from patients’ homes to their treatment centers. In pancreatic cancer, the intermediate category for distance in deciles ranged from 10 km to 89 km, whereas in rectal cancer, it was 10 km to 30 km. Figure 2 and Figure 3 show the distribution of cases according to the patients’ home addresses and their reference centers for pancreatic and rectal cancer. In the first case, reference centers were present throughout the region, except for the Pyrenees area, where patients must travel further for treatment. A comparison of Figure 3a (all cases of rectal cancer) and Figure 3b (90th percentile and higher) showed that the 10% of rectal cancer patients who lived furthest away from their treatment centers were distributed evenly throughout the region. In pancreatic cancer, the 10% of patients travelling the longest distances were treated in just three reference centers. In patients with both pathologies, no significant differences were observed in tumor stage according to distance category (Table 2).

On the other hand, patients’ physical statuses did show significant differences according to distance, with a greater percentage of unknown values in the intermediate category in the case of rectal cancer (Table 2). In relation to income categories, no significant differences were observed in either pathology (Table 2). However, the median distance was greater in patients with low compared to medium–high income in both pathologies (pancreatic cancer: median 8.26 km (IQR 35.68) versus 5.4 km (IQR 18.14); rectal: 4.52 km (IQR 10.18) versus 3.94 km (IQR 7.44); Appendix A). Similarly, in rectal cancer, no significant differences were observed in the distribution of the distance category according to age group, but when distance was analyzed as a continuous variable, it was negatively correlated with age group: the older the patients, the less distance they had to travel (from a median of 5.63 km (IQR 9.83) in people under 60 to 3.75 km (IQR 7.66) in those aged 80 years or older; *p* = 0.019). These differences were not observed in the case of pancreatic cancer (Appendix A).

### 3.1. Factors Influencing Survival in Patients with Pancreatic Cancer

Among the included cases of pancreatic cancer, five-year survival was 33.4%. Risk factors for mortality were low income and not having a textbook outcome (Table 3, Figure 4).

### 3.2. Factors That Influence Survival in Patients with Rectal Cancer

The five-year crude survival rate for rectal cancer was 67.5% (Figure 5). In the survival analysis adjusted for sex, age, ASA, and stage, patients who lived further from the 90th% cut-off had better survival than patients who lived less than 10 km from the hospital where the surgery was performed (Table 4). On the other hand, patients with a very low income level had a higher risk than those with medium–high income (Table 4, Figure 5). Not having a textbook outcome was also a risk factor (Table 4).

In the subanalysis of the 237 rectal cancer patients and the 168 pancreatic cancer patients operated on in reference centers outside their health districts, we did not observe differences in five-year survival.

## 4. Discussion

In this population-based study, which included over 2000 cases of pancreatic and rectal cancer operated on in the Catalan public healthcare system, the results did not show that a longer distance from a patient’s home to a reference center was associated with worse survival. The usual adjustment variables (sex, age, tumor stage, and ASA grade) did not contribute to any relationship between distance and survival in pancreatic cancer. In rectal cancer, however, the 10% of patients who lived furthest from their reference centers showed better survival than those who lived within 10 km. This fact might be attributable to social determinants. Focusing solely on the quality of care, as measured by the achievement of a textbook outcome, this factor did have an impact on survival at five years for both cancer diseases. Regarding socioeconomic status, we observed an association between low income and worse survival. Specifically for rectal cancer, compared to patients with medium–high income, those with very low income had a 5.2-fold higher risk of dying, and those with a low income had a 1.3-fold higher risk. In the case of pancreatic cancer, low-income patients had a 1.5-fold higher risk compared to patients with middle–high income.

Evidence has shown how distance can impact the probability of undergoing a full range of cancer treatments and surviving [12,13,14,24], as well as how socioeconomic inequalities can amplify such differences [25,26]. In a review on the origins of socioeconomic inequalities in cancer survival, Woods et al. highlighted that, after adjusting for socioeconomic differences, distance was related to worse tumor stage at diagnosis and shorter survival [18]. Although previous studies have often taken place in regions with a much lower population density than Catalonia (for instance, Queensland (Australia) is 57 times larger and has 2.5 million fewer inhabitants), the centralization strategy implemented in 2012 disrupted a deeply rooted hospital culture in which centers grew naturally in services and technology, treating a wide range of oncological pathologies with negligible coordination between them. Centralization entailed an increase in the distance to reference centers for 42% and 12% of pancreatic and rectal cancer patients, respectively, but without any pre-established clinical coordination formulas or inclusion policies for patients with highs burden of comorbidity or in situations of poverty [17]. In fact, one of the most significant results of this study was the association between lower income and worse five-year survival, which is consistent with other previous studies on survival in pancreatic and rectal cancer [27,28]. The short-term results (i.e., measured as the achievement of a textbook outcome) were comparable between the income groups, suggesting that the health system manages to be equitable in the short term, but it cannot avoid the effect of social inequality on patient life expectancy.

On the other hand, different studies have related the quality of the hospitalization process, measured using the ‘textbook outcome’ endpoint, to medium- and long-term survival [21,22]. In our study, textbook outcomes were independently associated with greater five-year survival in both the rectum and the pancreas, always adjusting for variables such as age, staging, or ASA, which numerous studies have also related to survival.

Regarding the real distance from patients’ homes, our data showed that most patients did not have to travel far for their surgeries: in rectal cancer, 90% of patients were treated at a center less than 30 km away. In the case of pancreatic cancer, subject to a more pronounced centralization strategy, the distance at the 90th percentile was 89 km. In pancreatic cancer patients who lived beyond this distance, a cluster of longer journeys was observed, especially concentrated in people living in one specific region. Since the end of the study period, this situation has changed, with the accreditation of an additional reference center that has significantly shortened the distances. In any case, greater distance did not translate to worse survival in either of the two pathologies studied. On the contrary, an unexpected result of our study was that 10% of rectal cancer patients who live furthest from their reference centers had better survival than patients living within a radius of 10 km. This finding raised several hypotheses, such as selection or self-selection of the ‘best cases’ to refer, along with a tendency to not refer patients with the worst prognosis, as they are considered palliative. As the analysis was adjusted for age, sex, and staging, this hypothesis is unlikely. A post hoc analysis by subregion showed that these cases were geographically spread out.

Looking longitudinally at the impact of cancer diagnosis on survivors’ earnings offers an important avenue for future research. In our study, the level of income corresponded to the year of surgery, which may subsequently decrease. Oncological processes can accelerate situations of material deprivation (e.g., lower income and professional interruptions) and end up imposing health inequalities. For its part, the distance that patients must travel to reference centers is also an issue of scientific relevance to the extent that the EU policy framework considers the need to specify resources and patients in expert centers. As a matter of fact, Europe’s Beating Cancer Plan (EBCP) “aims to ensure that 90% of eligible cancer patients have access to National Comprehensive Cancer Centres (CCCs) by 2030” [29].

Among the strengths of this study, the income measure used was individual and not grouped by territory or postal code, which would have introduced a risk of bias. Likewise, the use of real-world data is an added value, since it was based on the evaluation of cancer services in real conditions in a population that included all risk groups. Another strength is that the study drew from the interprofessional work of different public institutions in which doctors, epidemiologists, cartographers, demographers, statisticians, and others collaborated.

The main limitation is that it did not include non-operated cases, for which a certain geographical distance could have contributed to the decision to not undergo a surgical intervention. Different studies have shown that the need to travel further as a result of centralization reduces rates of treatment utilization for patients and widens inequities for those less able to travel [30,31]. However, the real impact of centralization on the use of curative surgery must be performed with population-based data that allow calculating the rate of surgery among incident cases. To obtain this information, a population study should be carried out including all incident cases in the study period. Additionally, we did not include the approximately 10% of cases treated in the private healthcare system, nor did we consider patients receiving surgery in non-reference centers in order to avoid the variability of results linked to unauthorized centers [15]. Patients operated on in reference centers outside their health districts were also excluded in consonance with the objective of the study, which was to assess the impact of centralization (a policy that establishes centers and, therefore, the distance to travel for surgery). However, in a subanalysis no differences were observed between these patients and those who attended their reference centers.

Socioeconomic status, while measured at the individual level, may be prone to some measurement errors, arising from the inability of official social security data to capture income from the informal economy or household income (except in minors). This could lead to the misclassification of some individuals, especially women (who show lower participation in the labor market); however, our results did not show gender-based differences in the association between socioeconomic status and mortality [23]. In cases of individuals who file joint tax returns or individuals in the same household with different incomes, this assessment procedure may introduce bias; however, no better information was available. In addition, workers in the informal economy tend to be of lower SES and would, therefore, be correctly categorized in the lowest SES category, as they would not be reporting any income.

Furthermore, in the database, annual individual income was captured as a categorical variable rather than as a continuous measure. This limited our ability to define exposure categories; in particular, the medium income stratum was very broad and, likely, heterogeneous [23]. In addition, because the high-income category included very few individuals, these were included in the same category as the middle-income group.

## 5. Conclusions

In short, the policy of centralizing highly complex oncological pathologies meant that a significant proportion of patients were treated outside their district hospitals, but this change was not associated with worse health outcomes compared to other patients, including that for survival. The reform was equitable in nature, as reflected by the short-term outcomes (textbook outcome, hospitalization); however, it did not erase the significant impact of income level on medium-term survival in patients with both rectal and pancreatic cancer. Growing clinical complexity and subspecialization, molecular diagnosis, and a policy context marked by pressures toward centralization will condition how health services are articulated to avoid health inequalities along the key planning axes in the coming years.

## Figures and Tables

**Figure 1 ijerph-19-08814-f001:**
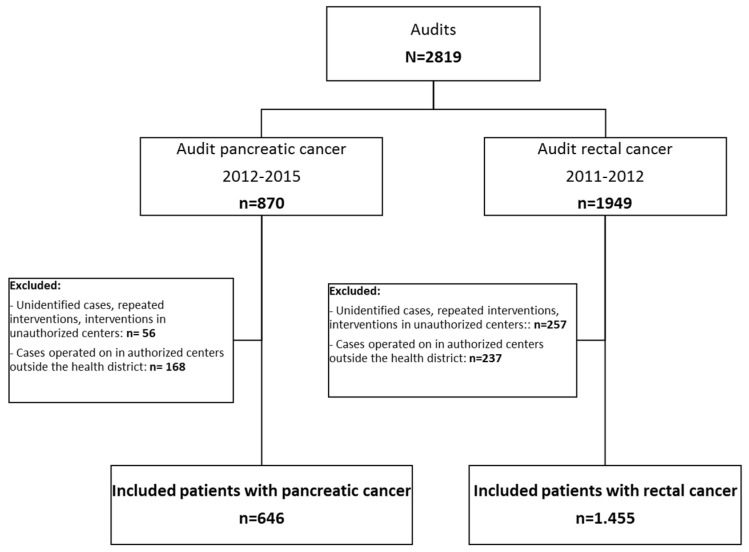
Study flow chart.

**Figure 2 ijerph-19-08814-f002:**
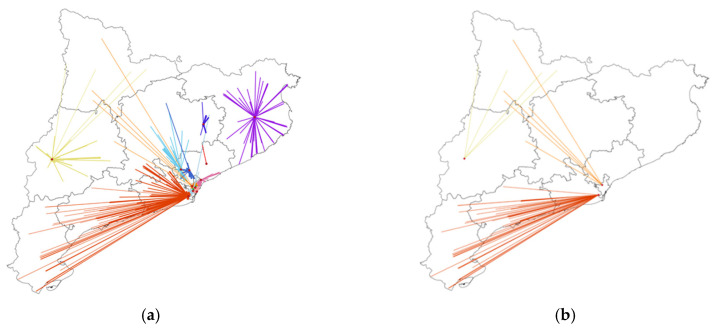
Distribution of pancreatic cancer cases according to locations of patients’ homes and surgical centers in Catalonia. (**a**) All cases of pancreatic cancer (N = 646). (**b**) Selection of cases with the longest distances (>90th%) between the patients’ homes and the reference centers (N = 64). Each color represents a different reference center.

**Figure 3 ijerph-19-08814-f003:**
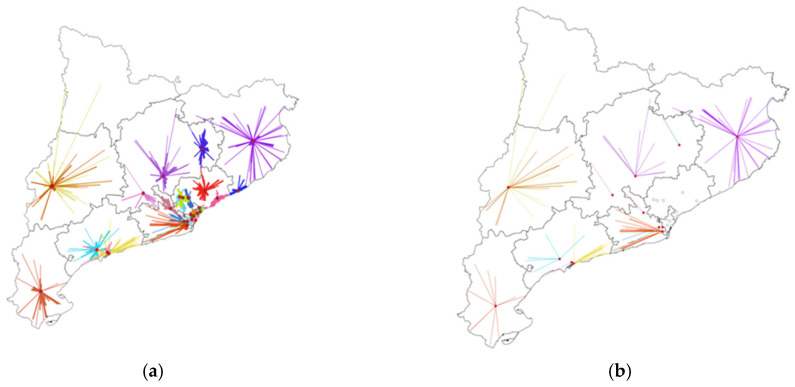
Distribution of rectal cancer cases according to locations of patients’ homes and surgical centers in Catalonia. (**a**) All cases of rectal cancer (N = 1455). (**b**) Selection of cases with the longest distances (>90th%) between the patients’ homes and the reference centers (N = 142). Each color represents a different reference center.

**Figure 4 ijerph-19-08814-f004:**
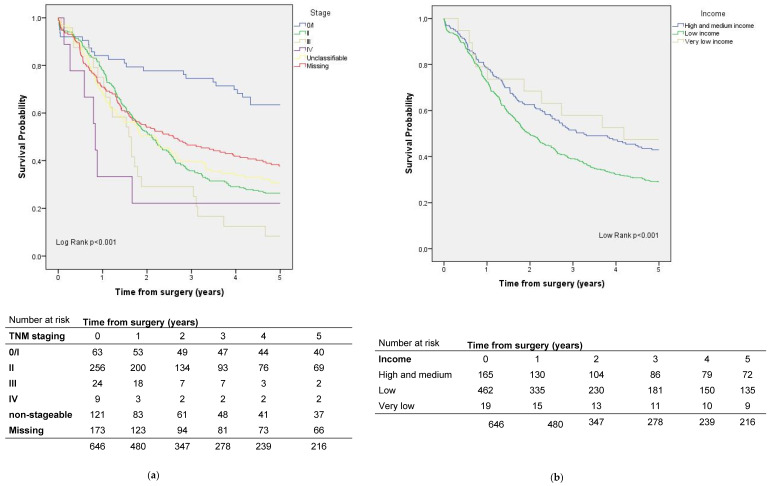
Survival by (**a**) stage and (**b**) socioeconomic status in pancreatic cancer patients.

**Figure 5 ijerph-19-08814-f005:**
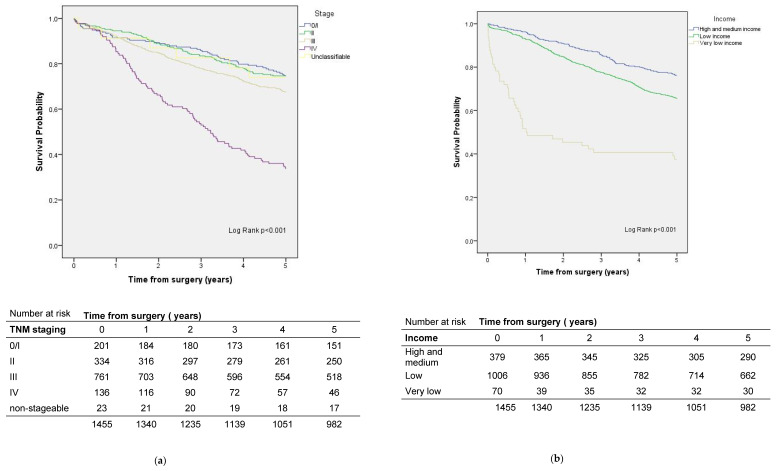
Survival by (**a**) stage and (**b**) socioeconomic status in rectal cancer patients.

**Table 1 ijerph-19-08814-t001:** Baseline characteristics of cohort by type of cancer.

Variables	Pancreatic Cancer (N = 646)		Rectal Cancer (N = 1455)	
	n	%	n	%
Sex				
Male	379	58.7	947	65.1
Female	267	41.3	508	34.9
Age				
Median (IQR)	69.0 (14.0)		69.6 (16.9)	
<60 years	141	21.8	311	21.4
60–69 years	206	31.9	423	29.1
70–79 years	227	35.1	459	31.5
≥80 years	72	11.1	262	18.0
Physical status score				
ASA I	17	2.6	89	6.1
ASA II	168	26.0	738	50.7
ASA III	162	25.1	486	33.4
ASA IV	13	2.0	49	3.4
Missing	286	44.3	93	6.4
TNM staging				
0/I	63	9.8	201	13.8
II	256	39.6	334	23.0
III	24	3.7	761	52.3
IV	9	1.4	136	9.3
Non-stageable	121	18.7	23	1.6
Missing	173	26.8	0	0.0
Income				
High–medium	165	25.5	379	26.0
Low	462	71.5	1006	69.1
Very low	19	2.9	70	4.8
Distance from patient residence to reference center (km)				
Median (IQR) Range	7.00 (32.83) 0.53–246.07		4.37 (8.89) 0.18–134.86	
0–10 km	380	58.8	1040	71.5
>10 km to < 90th% *	202	31.3	273	18.8
≥90th% *	64	9.9	142	9.8
District hospital				
Non-reference center	273	42.3	183	12.6
Reference center	373	57.7	1272	87.4

ASA: American Society of Anaesthesiologists; IQR: Interquartile range. * 90th%: pancreatic cancer, 89.14 km; rectal cancer, 29.70 km. Missing: no data found.

**Table 2 ijerph-19-08814-t002:** Baseline characteristics according to distance between patients’ homes and treatment centers.

		Pancreatic Cancer (N = 646) Rectal Cancer (N = 1455)											
		0–10 km		>10 km to <90th%		≥90th%		0–10 km		>10 km to <90th%		≥90th%	
		n	%	n	%	n	%	n	%	n	%	n	%
**Age**	<60 years	84	22.1	42	20.8	15	23.4	208	20.0	70	25.6	33	23.2
	60–69 years	113	29.7	70	34.7	23	35.9	300	28.8	84	30.8	39	27.5
	70–79 years	134	35.3	74	36.6	19	29.7	333	32.0	76	27.8	50	35.2
	≥80 years	49	12.9	16	7.9	7	10.9	199	19.1	43	15.8	20	14.1
	*p-value*	0.52						0.21					
**Physical status score**	ASA I	9	2.4	5	2.5	3	4.7	62	6.0	19	7.0	8	5.6
	ASA II	105	27.6	45	22.3	18	28.1	540	51.9	131	48.0	67	47.2
	ASA III	86	22.6	64	31.7	12	18.8	346	33.3	88	32.2	52	36.6
	ASA IV	9	2.4	3	1.5	1	1.6	38	3.7	5	1.8	6	4.2
	ASA V	0	0.0	0	0.0	0	0.0	0	0.0	0	0.0	0	0.0
	Missing	171	45.0	85	42.1	30	46.9	54	**5.2**	30	**11.0**	9	6.3
	*p-value*	0.33						0.044 *					
**Tumor stage**	0/I	46	12.1	12	5.9	5	7.8	151	14.5	30	11.0	20	14.1
	II	142	37.4	87	43.1	27	42.2	244	23.5	62	22.7	28	19.7
	III	16	4.2	8	4.0	0	0.0	538	51.7	147	53.8	76	53.5
	IV	7	1.8	1	0.5	1	1.6	87	8.4	31	11.4	18	12.7
	Non-stageable	69	18.2	41	20.3	11	17.2	20	1.9	3	1.1	0	0.0
	Missing	100	26.3	53	26.2	20	31.3	0	0.0	0	0.0	0	0.0
	*p-value*	0.29						0.24					
**Income**	High-medium	108	28.4	41	20.3	16	25.0	291	28.0	63	23.1	25	17.6
	Low	258	67.9	158	78.2	46	71.9	697	67.0	199	72.9	110	77.5
	Very low	14	3.7	3	1.5	2	3.1	52	5.0	11	4.0	7	4.9
	*p-value*	0.11						0.052					
**Total**		380	100.0	202	100.0	64	100.0	1040	100.0	273	100.0	142	100.0

ASA: American Society of Anaesthesiologists. 90th%: pancreatic cancer, 89.14 km; rectal cancer, 29.70 km. *p*-*values* below 0.05 (two-sided) were considered to indicate statistical significance. Chi-square test.* Statistically significant.

**Table 3 ijerph-19-08814-t003:** Multivariate survival analysis of pancreatic cancer patients at 5 years of follow-up.

		Univariate Analysis			Multivariate Analysis		
Variables	n	HR	95% CI	*p*-Value	HR	95% CI	*p*-Value
Sex							
Male	379	1			1		
Female	267	0.97	0.80, 1.77	0.79	0.87	0.71, 1.06	0.71
Age							
<60 years	141	1		<0.001	1		0.008
60–69 years	206	1.52	1.15, 2.02	0.004	1.37	1.02, 1.84	0.034
70–79 years	227	1.75	1.33, 2.31	<0.001	1.53	1.15, 2.05	0.004
≥80 years	72	2.02	1.42, 2.86	<0.001	1.82	1.25, 2.65	0.002
Physical status score							
ASA I	17	1		0.014	1		0.067
ASA II	168	2.18	0.96, 4.97	0.064	1.75	0.76, 4.04	0.19
ASA III	162	2.97	1.31, 6.76	0.009	2.02	0.87, 4.70	0.10
ASA IV	13	3.21	1.17, 8.85	0.024	2.31	0.82, 6.48	0.11
Missing	286	2.85	1.26, 6.42	0.012	2.93	1.20, 7.19	0.019
TNM staging							
0/I	63	1		<0.001	1		<0.001
II	256	2.78	1.80, 4.30	<0.001	2.84	1.82, 4.42	<0.001
III	24	4.22	2.35, 7.59	<0.001	3.37	1.86, 6.12	<0.001
IV	9	4.59	1.97, 10.71	<0.001	4.52	1.80, 11.33	0.001
Non-stageable	121	2.68	1.69, 4.25	<0.001	1.70	0.96, 3.03	0.070
Missing	173	2.24	1.43, 3.51	<0.001	1.60	0.94, 2.72	0.083
Textbook outcome							
Yes	284	1			1		
No	362	1.48	1.22, 1.79	<0.001	1.344	1.10, 1.64	0.004
Distance from patient home to reference center							
0–10 km	380	1		0.90	1		0.92
11 km to <90th%	202	1.04	0.84, 1.28	0.74	1.05	0.80, 1.39	0.71
≥90th%	64	0.96	0.70, 1.33	0.81	1.08	0.72, 1.61	0.73
Income							
High–medium	165	1		0.003	1		0.003
Low	462	1.45	1.15, 1.82	0.002	1.47	1.15, 1.87	0.002
Very low	19	0.87	0.45, 1.67	0.678	0.84	0.43, 1.63	0.60
District center							
Non-reference center	273	1			1		
Reference center	373	1.03	0.85, 1.25	0.75	1.14	0.86, 1.50	0.362

CI: confidence interval; HR: hazard ratio. *p*-values below 0.05 (two-sided) were considered to indicate statistical significance.

**Table 4 ijerph-19-08814-t004:** Multivariate survival analysis of rectal cancer patients at 5 years of follow-up.

		Univariate Analysis			Multivariate Analysis		
Variables	n	HR	95% CI	*p*-Value	HR	95% CI	*p*-Value
Sex							
Male	947	1			1		
Female	508	0.96	0.79, 1.16	0.64	0.95	0.78, 1.15	0.59
Age							
<60 years	311	1		<0.001	1		<0.001
60–69 years	423	1.07	0.78, 1.45	0.69	1.17	0.86, 1.61	0.32
70–79 years	459	1.79	1.35, 2.37	<0.001	1.85	1.38, 2.49	<0.001
≥80 years	262	2.87	2.15, 3.84	<0.001	3.33	2.43, 4.57	<0.001
Physical status score							
ASA I	89	1		<0.001	1		<0.001
ASA II	738	1.43	0.86, 2.38	0.17	1.15	0.68, 1.95	0.60
ASA III	486	2.74	1.65, 4.56	<0.001	1.96	1.16, 3.34	0.013
ASA IV	49	5.28	2.90, 9.63	<0.001	3.73	1.98, 7.01	<0.001
Missing	93	3.15	1.78, 5.60	<0.001	2.35	1.30, 4.26	0.005
TNM staging							
0/I	201	1		<0.001	1		<0.001
II	334	1.01	0.71, 1.43	0.97	1.04	0.73, 1.48	0.81
III	761	1.36	1.00, 1.85	0.047	1.72	1.26, 2.34	0.001
IV	136	3.61	2.55, 5.10	<0.001	4.77	3.34, 6.81	<0.001
Non-stageable	23	1.05	0.45, 2.45	0.91	0.85	0.36, 1.99	0.70
Textbook outcome							
Yes	529	1			1		
No	926	1.58	1.30, 1.93	<0.001	1.46	1.20, 1.79	<0.001
Distance from patient home to reference center							
0–10 km	1040	1		0.70	1		0.044
11 km to <90th%	273	1.09	0.87, 1.37	0.47	1.08	0.84, 1.38	0.55
≥90th%	142	0.95	0.70, 1.30	0.75	0.67	0.46, 0.97	0.034
Income							
High–medium	379	1		<0.001	1		<0.001
Low	1006	1.55	1.23, 1.96	<0.001	1.28	1.00, 1.63	0.047
Very low	70	4.69	3.23, 6.81	<0.001	5.24	3.58, 7.67	<0.001
District center							
Non-reference center	183	1			1		
Reference center	1272	0.78	0.61, 1.01	0.058	0.76	0.55, 1.04	0.081

CI: confidence interval; HR: hazard ratio. *p*-values below 0.05 (two-sided) were considered to indicate statistical significance.

## Data Availability

Data are available on request.

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
