# Peer review of "Socioeconomic Status and Distance to Reference Centers for Complex Cancer Diseases: A Source of Health Inequalities? A Population Cohort Study Based on Catalonia (Spain)"

_ijerph, 2022, doi:10.3390/ijerph19148814_

Round 1

Reviewer 1 Report

Overall, the article is organized, easy to follow, and a diverse array of credible sources are referenced. The article aims and successfully investigates whether geographical location and socioeconomic status relate to survival in rectal and pancreatic cancer patients. Clear thought process on the analysis of data and the selection criteria were exhibited, however, clarity and acknowledging study limitations are important, hence a few suggestions are outlined for the authors’ consideration.

1.     Similar studies investigating the impact of distance on rectal and pancreatic cancer survival from the Discussion can be moved to the Introduction to provide logic for the need of this research study.

2.     Should include the limitations and biases of including self reported income?

3.     What were the significance and implications of using the “textbook outcome” other than including it as a single indicator? Does it indicate better quality of care overall? Explain why this parameter was used as an exclusion criterion. 

4.     For Tables 1, 2 and 3, perhaps break up the large data tables and summarize key data trends in smaller tables for readability and easier comparisons for the readers. 

5.     Explain the meaning of using different colours in Figures 2 and 3 (Eg. Each colour representing a different reference centre)

6.     Studies in countries with a similar population pattern as Catalonia can better validate the current study’s results. 

7.     Did not propose specific areas of future research to investigate the relationship between distance and cancer survival or perhaps health outcomes of other diseases

8.     Could propose more alternative hypothesises on unexpected data. For instance, an alternative hypothesis for a better survival rate in the 10% of patients who live the furthest could be attributed to social influences (Eg. Better air quality and less environmental stressors).

9.     I wonder if increasing the distance intervals could make a difference when analyzing data. Perhaps the greatest distance travelled is not extremely far for the furthest patients to travel. If so it could account for the lack of significance between distance and health outcome, and pose as a future research interest.

Author Response

Response to Reviewer 1 Comments

  1. Similar studies investigating the impact of distance on rectal and pancreatic cancer survival from the Discussion can be moved to the Introduction to provide logic for the need of this research study.

Thank you very much for your suggestion. We have added the following to the introduction (line 61): “In fact different studies have shown how geographical distance can impact the probability of undergoing a full range of cancer treatments and the chances of survival. (Dejardin et al., Crawford et al., Baade et al. others)     

  1. Should include the limitations and biases of including self-reported income?

Thank you for the comment. However, “self-reported” refers to individual income tax returns.

The last paragraph of the Discussion deals with the limitations and biases associated with this variable. In any case, we have added in the discussion ( last paragraph): “In the case of individuals who file joint tax returns or individuals in the same household with different incomes, this assessment procedure may introduce bias; however, no better information is available. Workers in the informal economy tend to be of lower SES and would therefore be correctly categorized in the lowest SES category as they will not be reporting any income.”

We also have added in the study design section: ”Individuals are assigned to a particular medication copayment group on the basis of declared income on individual tax returns.”

  1. What were the significance and implications of using the “textbook outcome” other than including it as a single indicator? Does it indicate better quality of care overall? Explain why this parameter was used as an exclusion criterion. 

To obtain a simple, systematic and summarized approximation of the quality of care, avoiding the traditional "siloed" outcome metrics, with which we could analyze the potential impact of the distance to the surgical centre while taking into account the variability of clinical practice, we used the Dutch model to summarize quality of care in a single composite indicator, the textbook outcome. This combination of multiple indicators that define a perfect hospitalization has been used mainly to study the variability between centres in terms of hospital performance in complex surgery. In addition to the quality of the hospitalization process, textbook outcomes have been related to medium- to long-term survival in different studies.

This was not an exclusion criterion (those that do not meet the outcome textbook are not excluded) but rather an adjustment variable.

- Kolfschoten NE, Kievit J, Gooiker GA, van Leersum NJ, Snijders HS, Eddes EH, Tollenaar RA, Wouters MW, Marang-van de Mheen PJ. Focusing on desired outcomes of care after colon cancer resections; hospital variations in 'textbook outcome'. Eur J Surg Oncol. 2013 Feb;39(2):156-63.

- Busweiler LA, Schouwenburg MG, van Berge Henegouwen MI, Kolfschoten NE, de Jong PC, Rozema T, Wijnhoven BP, van Hillegersberg R, Wouters MW, van Sandick JW; Dutch Upper Gastrointestinal Cancer Audit (DUCA) group. Textbook outcome as a composite measure in oesophagogastric cancer surgery. Br J Surg. 2017 May;104(6):742-750.

- Sweigert PJ, Eguia E, Baker MS, Paredes AZ, Tsilimigras DI, Dillhoff M, Ejaz A, Cloyd J, Tsung A, Pawlik TM. Assessment of textbook oncologic outcomes following pancreaticoduodenectomy for pancreatic adenocarcinoma. J Surg Oncol. 2020 May;121(6):936-944.

  1. For Tables 1, 2 and 3, perhaps break up the large data tables and summarize key data trends in smaller tables for readability and easier comparisons for the readers. 

Following the advice of the reviewer and with the aim of facilitating the reading of Tables 1 and 2, we have eliminated the columns for the total (Table 1), as this is not relevant. In Table 2, we eliminated the information on the total number of cases within each pathology, as it is already shown in Table 1 and was therefore redundant. As for Table 3 (pancreatic cancer), which in fact is the same as table 4 (rectal cancer), we decided to conserve the format and the information it contains, as we believe results of both the univariate and multivariate analyses are relevant, and concerning the outcomes and adjustment variables.

  1. Explain the meaning of using different colours in Figures 2 and 3 (Eg. Each colour representing a different reference centre)

The reviewer is correct: each colour represents a different reference centre. This clarification has been added to the map legend.

  1. Studies in countries with a similar population pattern as Catalonia can better validate the current study’s results..

Although we fully agree with the reviewer, we were unable to identify similar studies in countries or regions with a similar population pattern as Catalonia, such as any region of France, such as Rhônes-Alpes. Perhaps this could be the subject of a review.

  1. Did not propose specific areas of future research to investigate the relationship between distance and cancer survival or perhaps health outcomes of other diseases

We thank the reviewer for this comment. One area of future research would be the impact of distance on the use of curative surgery, using population data that allows calculation of the rate of surgery among incident cases in the study period. It has been added that ”However, the real impact of centralization on the use of curative surgery must be done with population-based data that allow calculating the rate of surgery among incident cases. To obtain this information, a population study should be carried out, including all incident cases in the study period. ( line 2 pg 13)

  1. Could propose more alternative hypothesises on unexpected data. For instance, an alternative hypothesis for a better survival rate in the 10% of patients who live the furthest could be attributed to social influences (Eg. Better air quality and less environmental stressors).

We appreciate the reviewer’s comment. We have added, “might be attributable to social determinants” (line 270)

  1. I wonder if increasing the distance intervals could make a difference when analyzing data. Perhaps the greatest distance travelled is not extremely far for the furthest patients to travel. If so it could account for the lack of significance between distance and health outcome, and pose as a future research interest.

The concept of distance is relative. As we discussed in the introduction, a 100 km journey does not mean the same thing in Catalonia and Australia. The intervals studied must make sense in the specific study setting. In our case, these intervals were decided based on their relationship with the population distribution in the metropolitan environment of large cities.

Reviewer 2 Report

Very Respected Authors.

After reading your manuscript I have some suggestions.

I could not see the type of the manuscript. Is this an original research or a review?  The whole manuscript is written as an original manuscript, but the Abstract  does not have the same parts as the manuscript.  If this original research then the Abstract has to be structured as the whole manuscript (Introduction Objective, Methods...).

Conclusion is missing in the manuscript. 

At the end of the manuscript stands:  In short, the policy of centralizing highly complex oncological pathologies meant that a significant proportion of patients were treated outside their district hospital, but this change was not associated with worse health outcomes compared to the other patients, including for survival. The reform was equitable in nature, as reflected by short-term outcomes (textbook outcome, hospitalization); however, it did not erase the significant impact of income level on medium-term survival in patients with both rectal and pancreatic cancer. Growing clinical complexity and sub-specialization, molecular diagnosis, and a policy context marked by pressures towards centralization will condition how health services are articulated to avoid health inequalities along the key planning axes in the coming years. 

I suggest for it to be written as Conclusion and then this text will be suitable.

Author Response

  1. I could not see the type of the manuscript. Is this an original research or a review?  The whole manuscript is written as an original manuscript, but the Abstract  does not have the same parts as the manuscript.  If this original research then the Abstract has to be structured as the whole manuscript (Introduction Objective, Methods...).

We thank the reviewer for his comments, The present paper is an original research manuscript, named "article" in IJERPH. Regarding the abstract, among the instructions for authors it says " The abstract should be a single paragraph and should follow the style of structured abstracts, but without headings: 1) Background: Place the question addressed in a broad context and highlight the purpose of the study; 2) Methods: Describe briefly the main methods or treatments applied. Include any relevant preregistration numbers, and species and strains of any animals used. 3) Results: Summarize the article's main findings; and 4) Conclusion: Indicate the main conclusions or interpretations. We have followed such instructions.

2. Conclusion is missing in the manuscript. 

At the end of the manuscript stands:  In short, the policy of centralizing highly complex oncological pathologies meant that a significant proportion of patients were treated outside their district hospital, but this change was not associated with worse health outcomes compared to the other patients, including for survival. The reform was equitable in nature, as reflected by short-term outcomes (textbook outcome, hospitalization); however, it did not erase the significant impact of income level on medium-term survival in patients with both rectal and pancreatic cancer. Growing clinical complexity and sub-specialization, molecular diagnosis, and a policy context marked by pressures towards centralization will condition how health services are articulated to avoid health inequalities along the key planning axes in the coming years. 

I suggest for it to be written as Conclusion and then this text will be suitable.

We have followed the suggestion of the reviewer.

Reviewer 3 Report

Thank you for a the opportunity to review a well constructed and presented manuscript. A couple of minor comments on the manuscript but overall I have no real suggestions for alteration but have one observation.

In the discussion on page 12 of the manuscript the point of exclusion of non-operated cases is raised. There are two issues with what is presented. 1. Is that stable numbers is not a good proxy for no impact as overall incidence of cancer should be rising as population ages and grows. Thus this should really be an assessment based on population data not operation numbers. Related in work done in NSW Australia we found that higher volume centres for pancreatic and oesophageal surgery operated on a higher proportion of patients presenting in their geographic region than in low volume centres - suggesting the greater the expertise of the team the more complex the surgery attempted and this did not result in higher complications or poorer survival. We felt that there was a level of nihilism in centres operating on a lower proportion of presenting patients and I would be concerned that as centralisation occurs these patients are not referred on. I am sure you have the data sets to check this and would encourage you to to consider this paragraph from these angles and set some thinking about what sort of follow up analysis might be warranted. 

Author Response

Response to Reviewer 3 Comments

  1. Is that stable numbers is not a good proxy for no impact as overall incidence of cancer should be rising as population ages and grows. Thus this should really be an assessment based on population data not operation numbers.

Certainly, the real impact of centralization on the use of curative surgery must be analysed through population data that allow calculating the rate of surgery among incident cases. However, we do not have such data. To do a population-based study of rectal cancer, it is necessary to use the information from the two population registries that exist in Catalonia and cover 2 of the 4 provinces (Girona and Tarragona). On the other hand, it is worth remembering that the framework of this study is that of evaluating the impact of centralization on patients undergoing surgery, and the main source of data is the hospital discharge database, which ensures the identification of surgical patients. The main limitation is that we do not have data on non-surgical patients.

We have deleted, “Our results suggest that this is not the case in Catalonia...” and added that “However, the real impact of centralization on the use of curative surgery must be done with population-based data that allow calculating the rate of surgery among incident cases. To obtain this information, a population study should be carried out, including all incident cases in the study period.”

  1. Related in work done in NSW Australia we found that higher volume centres for pancreatic and oesophageal surgery operated on a higher proportion of patients presenting in their geographic region than in low volume centres - suggesting the greater the expertise of the team the more complex the surgery attempted and this did not result in higher complications or poorer survival. We felt that there was a level of nihilism in centres operating on a lower proportion of presenting patients and I would be concerned that as centralisation occurs these patients are not referred on. I am sure you have the data sets to check this and would encourage you to to consider this paragraph from these angles and set some thinking about what sort of follow up analysis might be warranted. 

Again, we do not have population data or data from patients who did not undergo surgery. Therefore, we cannot analyze whether some unauthorized centres do not refer their patients.

Comments in the text:

  1. are their thresholds for the two cancers looked at in this paper as that would be good to include. Also number of centres (you might say that later)

We have introduced the thresholds: 10 cases per year in pancreatic surgery and 11 per year in rectal cancer.

  1. Definition of Income: might be useful to explain this a little further. How many levels and corresponding to what income?

The paragraph beginning on line 152 includes the income categories:

High SES: annual income >EUR 100,000 /year

Medium SES: annual income EUR 18,000-100,000/year

Low SES: anual income < EUR 18,000/year

Very low SES ( individuals receiving welfare support from the government)

We have added: “Individuals are assigned to a particular medication copayment group on the basis of declared income on individual tax returns.”

Study design

  1. Can you clarify that no patients are treated outside of reference centres. In Australia consolidation efforts happen but it is difficult to stop individual surgeons performing various operations

It took two years for regional hospitals to comply with the new regulations, reflecting the complexities (including financial constraints) that this kind of policy measure inevitably entails. Since then, the adherence to the centralization decree is almost complete. In addition, about 10% of cases are operated on in private centres.

Prades J, Manchon-Walsh P, Solà J, Espinàs JA, Guarga A, Borras JM. (2016). Improving clinical outcomes through

centralization of rectal cancer surgery and clinical audit: a mixed-methods assessment. European Journal of Public Health, 26(4), 538-542.

  1. the first bit answers my question above but how many excluded? 43 for pancreatic cancer and 238 in rectal cancer excluded to be treated in unauthorized centres.

Second part - assume this means each area is designated to a reference centre but people move to other centres for reasons such as having relatives to stay with. Not sure why they are excluded? Distance relationship issue I guess?

Indeed, these are what we called "by-passers", that is, patients who decided not to be treated at their reference center as the regulation dictates and instead travelled a greater distance to go to another reference centre that does not correspond to their treatment area.

A supplementary analysis has been carried out including these patients, with similar results.

  1. Surgery for pancreatic cancer took place in 12 centres and for rectal cancer, in 27, reflecting differences in the centralization strategies between pathologies that condition the distance from patients’ homes to their treatment centre. And presumably differences in incidence?

Yes, of course, it is a criterion that was taken into account at the time.

  1. Line 219 missing 'to' medium ...

Thank you, corrected

Discussion

  1. We deleted, “Our results suggest that this is not the case in Catalonia: first of all, the number of rectal cancer cases undergoing surgery remained stable before (2005- 2007, N = 1831) and after (2011-2012, N = 1949) centralization, and secondly, mean patient age actually increased between these two periods, from 68.1 to 69.6 years. …”  as this not might not signify stability, as it depends on incidence.

We added, “However, the real impact of centralization on the use of curative surgery must be done with population-based data that allow calculating the rate of surgery among incident cases. To obtain this information, a population study should be carried out, including all incident cases in the study period.”

Reviewer 4 Report

Authors  in their manuscript entitled “Socioeconomic Status and Distance to Reference Centres for Complex cancer diseases: a source of health inequalities ? a population cohort study based obn Catalonia (Spain)”  describe a cohort of 2000 cancer patients operated on in the Catalan health system. One of the objectives of the study was to demonstrate if the longer distance from the patient’s home to the Center could impact on the survival. They find that was an association with the income status and worse survival but not with the patients’ home distance. The manuscript is interesting . I recommend for the publication.

Author Response

Thank you very much for your comments.

Sincerely,

Round 2

Reviewer 1 Report

Thank you for your revision. All of my concerns have been addressed.